# Synergetic Roles of Mangrove Vegetation on Sediment Accretion in Coastal Mangrove Plantations in Central Thailand

Sarawan Hongwiset [1,2], Chadtip Rodtassana [1,*], Sasitorn Poungparn [1], Suthathip Umnouysin [3] and Vilanee Suchewaboripont [4]

1    Department of Botany, Faculty of Science, Chulalongkorn University, Bangkok 10330, Thailand
2    Interdisciplinary Program of Environmental Science, Graduate School, Chulalongkorn University, Bangkok 10330, Thailand
3    Department of Biology, Faculty of Science, Silpakorn University, Nakhon Pathom 73000, Thailand
4    The Institute for the Promotion of Teaching Science and Technology, Bangkok 10110, Thailand
*    Correspondence: chadtip.r@chula.ac.th; Tel.: +66-2-218-6156

**Abstract:** Mangrove plantations can potentially restore the vegetation structures protecting coastal areas. In *Avicennia alba* plantations in central Thailand, we investigated the vegetation structures (trees, seedlings, pneumatophores, and belowground roots), sedimentation rates, and elevation changes over one year across the shore. The results showed a heterogeneous vegetation structure with an increasing tree basal area (BA) and seedling density towards the interior. The pneumatophore quantitative characteristics decreased towards the interior in association with the topographic gradient and inundation period. The sedimentation rates, which were greater in the plantation than on the mudflat, showed a negative correlation with the height, total surface area, and total volume of the pneumatophores. This indicates that the pneumatophores facilitated the transportation of fine sediments into the interior. Our results suggest that an optimal complexity of the aboveground vegetation structure might enhance the sedimentation rates. According to the gradient of tree BA and seedling density, the fine root density increased towards the interior. The monthly elevation changes in the plantation showed less fluctuation than those on the mudflat. The net elevation changes indicated sediment accretion within the plantation and erosion on the mudflat. Our results demonstrate the synergetic roles of mangrove plantations in which the aboveground structures facilitate sediment redeposition and the belowground roots stabilize sediment accretion in coastal areas.

**Keywords:** *Avicennia alba*; pneumatophore; fine root; sedimentation rate; elevation change; mangrove plantation

## 1. Introduction

Coastal mangrove ecosystems have been particularly susceptible to environmental changes and unusual extreme events caused by climate change [1–3]. Additionally, the accommodating areas of coastal mangrove ecosystems have been restricted due to limited landward expansion resulting from urban and infrastructure development [4] and limited seaward expansion due to a rise in the sea level [5]. Mangrove ecosystems provide significant ecosystem services, such as acting as global carbon stocks and exchanges as a component of blue carbon [6,7] and sediment accretion [8]. Restricting areas for mangrove ecosystems leads to a failure to provide the abovementioned functions due to threats from natural disturbances and anthropogenic activities.

Mangrove plantations are one of the conventional approaches to rehabilitating degraded or lost mangrove forests that mainly aim to restore vegetation structure not only for carbon sequestration but also for coastal stabilization and environmental mitigation [9,10]. Although most mangrove plantations are considered low-diversity plantings or monospecific plantations [9], successful mangrove rehabilitation likely involves the storage of

carbon in tree biomass and consequently enhances stem density faster than natural regeneration [11]. Therefore, mangrove plantations could be acknowledged as providing other ecosystem services, including coastal protection [12] and sediment accretion [13,14] in coastal areas.

Several studies have reported that mangroves facilitate sedimentation processes in coastal areas in relation to their vegetation structures, especially the aerial roots of mangroves, which enhance sediment trapping and accretion [15,16]. Moreover, the high density of mangrove seedlings induces high sediment accretion in plantations [17,18]. The aboveground structures, including mangrove stems, seedlings, and aerial roots, cause sediment retention by reducing the velocity of waves and increasing local turbulence [19,20]. Belowground roots also play an important role in sediment accretion and surface elevation via subsurface processes [16,21]. According to a recent study by Karimi et al. [22] in an Iranian mangrove forest, the higher root length density of *A. marina* in the seaward area caused a lower soil detachment ratio and less erosion compared to the landward area. This suggests that the presence of mangrove roots may help stabilize the shoreline. However, there have been no concurrent observations in mangrove forests regarding the roles of the aboveground and belowground components in assisting sedimentation processes.

Environmental gradients (i.e., topography, inundation period) across mangrove forests cause heterogeneous vegetation structures that influence different sedimentation patterns within mangrove forests [23,24]. Aerial roots also show morphological plasticity depending on environmental gradients [16]. For example, the pneumatophores of *Avicennia* change their morphological characteristics by increasing the pneumatophore size to sustain their function of gas exchange under longer inundation periods in shore areas [25,26]. An experiment in a laboratory using mangrove root models showed that the high density of aerial roots led to a high level of sediment accumulation posterior to the roots [27]. However, some field observations have reported that the presence of pneumatophores also causes erosion around them [28]. The erosion occurred temporarily, and there might be other equivalent mechanisms that influence erosional and sedimentation processes through the spatial heterogeneity of the aerial roots and other vegetation components across mangrove forests.

The mangrove forests in Samut Prakarn province in central Thailand were previously disturbed by human activities, such as industrial and urban development [29]. Subsequently, a mangrove restoration project was initiated in 2005 by the Quartermaster Department of the Royal Thai Army (hereafter RTA) along with the public and private sectors at the Bangpu Recreation Center in Samut Prakarn province (approximately 15 km from Bangkok). The restoration programs were regularly organized in various designated areas along the shoreline at the Bangpu Recreation Center every year. The RTA (government sector) and the public and private sectors collaborated on several restoration activities to increase public awareness of mangrove forests, which is known as corporate social responsibility (CSR).

Therefore, we are interested in the sediment accretion occurring in this mangrove plantation, focusing on the influence of the restored vegetation structure. This study aimed to investigate: (1) the roles of mangrove vegetation, including its aboveground components (trees, seedlings, and pneumatophores) and belowground roots, on sediment accretion in comparison to the non-vegetated areas (mudflats); and (2) the variation in quantitative characteristics of pneumatophores along the cross-shore transects in this coastal mangrove plantation. We hypothesized that the presence of mangrove vegetation would enhance the sediment accretion in mangrove plantations in association with the characteristics of the vegetation structure. We proposed the vegetation structure has synergetic roles in mangrove plantations which provide ecosystem services that facilitate coastal stabilization and environmental mitigation via sedimentation processes.

## 2. Materials and Methods

### 2.1. Study Site

The study site is a mangrove plantation in Samut Prakarn province in central Thailand (13°31′ N, 100°39′ E), which is located at an elevation of 4 m from the mean sea level at the Bangpu Recreation Center on the eastern shore of the Chao Phraya River mouth (Figure 1a). Mangrove seedlings from several origins and mangrove nurseries in central, eastern, and southern Thailand were planted in plantations and included *Avicennia alba*, *Rhizophora mucronata*, *R. apiculata*, and *Sonneratia caseolaris*. The seedlings used for planting were approximately 7–8 months old (for the *Rhizophora* seedlings) and more than 6–12 months old (for the *Avicennia* and *Sonneratia* seedlings). At the beginning of planting, each seedling was planted systematically at an interval of 1 m. However, the dominant tree at the Bangpu mangrove plantations is *A. alba*, which is considered the fast-growing pioneer mangrove species colonizing new mudflats [30]. We conducted our research in mangrove plantations aged 8–10 years, as shown in Figure 1b. The tidal pattern is characterized as mixed semidiurnal [31], with tides ranging from 0 to 3.4 m (September 2020 to November 2021, the Hydrographic Department, Royal Thai Navy).

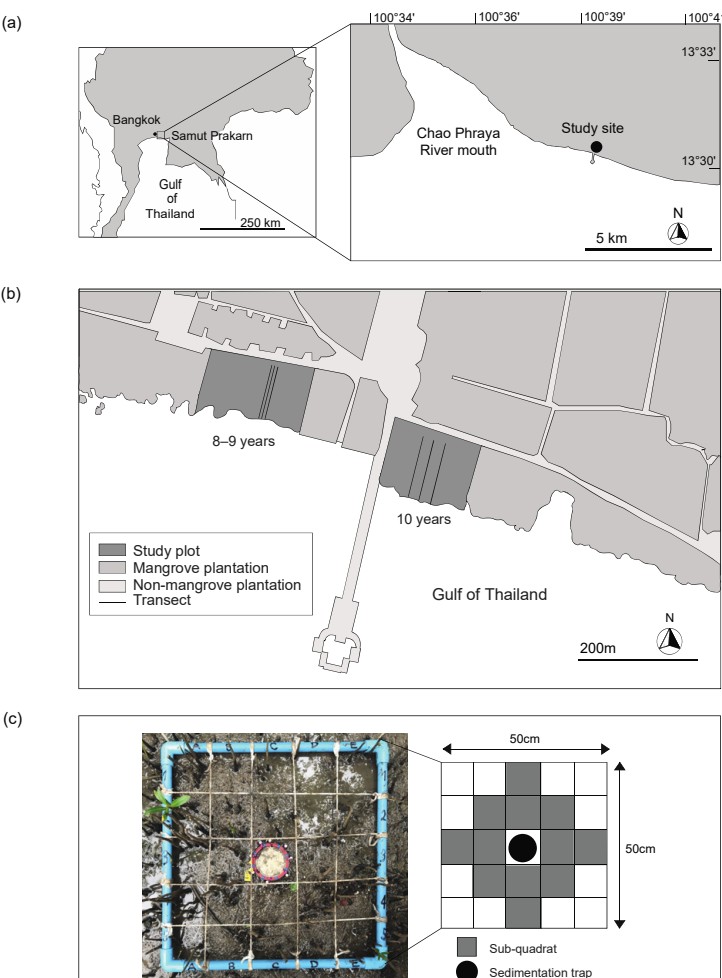

**Figure 1.** Location of the mangrove plantation in Samut Prakarn province in central Thailand. (**a**) study site on the eastern shore of the Chao Phraya River mouth; (**b**) study plots in the mangrove plantations; and (**c**) a quadrat for a study of pneumatophore characteristics and sedimentation rates.

The climatic data over 10 years from January 2009 to December 2019 were received from the Num Rong weather station, which has a long-term data record and is the nearest to the study site, located approximately 16 km away (Meteorological Department of Thailand). This study site experiences a tropical monsoon climate with a distinct rainy season (May to October) and dry season (November to April). The rainy season receives 82.5% of the annual rainfall, of which the annual rainfall is 1008 mm. The maximum air temperature is in May (30.5 °C), while the lowest is in January (26.8 °C), with an average temperature of 28.9 °C. The wind speed ranges from 10.9 to 19.9 km hour$^{-1}$.

During the one-year study period (September 2020 to August 2021), the climatic data were received from the Samut Prakan (Bang Pla) weather station, which is the nearest station to the study site, located approximately 11 km away (Meteorological Department of Thailand). The annual rainfall was 1512.5 mm, and the rainy season received 86% of the annual rainfall. The average air temperature was 28.6 °C, with a maximum in May 2021 and a minimum in January 2021. The wind speed varied within a range from 4.8 km hour$^{-1}$ recorded in October 2020 to 11.1 km hour$^{-1}$ recorded in July 2021 (Figure 2).

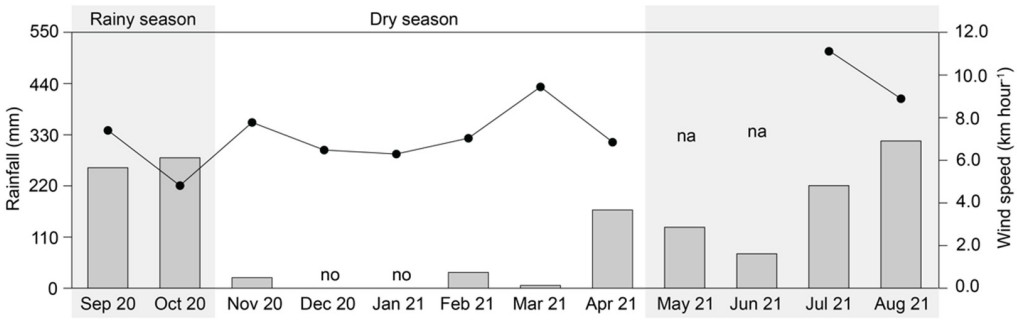

**Figure 2.** Climatic characteristics obtained from the Samut Prakan (Bang Pla) weather station, including monthly rainfall (bar graph) and wind speed (line graph) from September 2020 to August 2021 (Meteorological Department of Thailand); 'no' indicates no rainfall, and 'na' indicates no record.

### 2.2. Sampling Points

Six belt transects (with 4 m width) were set up from the edge of the 8–10-year-old mangrove plantation across a distance of 100 m towards the interior (Figure 1b). Each transect was spaced 10–20 m apart. The vegetation structure was investigated at every 10 m section along each transect. The systematic sampling points were designated at intervals of 0, 10, 30, 50, 70, and 90 m from the shore, and we examined the roots, sediment, and environmental factors at these sampling points. Moreover, a sampling point was set on the mudflat approximately 5 m from the fringe (0 m) of each transect to examine the sediment and environmental factors. For all transects, a quadrat of $50 \times 50$ cm$^2$ was set at each sampling point to study the quantitative characteristics of pneumatophores, the number of seedlings, and the sedimentation rates (Figure 1c).

### 2.3. Vegetation Structure

All trees in the belt transect with a diameter at breast height (DBH) greater than 4.5 cm were identified to species and measured for DBH in September 2020 and October 2021. Tree density was calculated from the total number of trees per area along the belt transect. The total stem basal area (BA) was calculated by the summation of the basal area of all the trees in the transect, assuming that the basal area of the tree was circular. For each transect, aboveground and belowground biomass was estimated by using the common allometric equations for mangrove species [32]. All seedlings (less than 1 m high) in a quadrat of $50 \times 50$ cm$^2$ at the sampling points (Figure 1c) along each transect were identified and counted to determine the seedling density every three months from September 2020 to September 2021.

### 2.4. Quantitative Characteristics of Pneumatophores and Belowground Roots

We divided the $50 \times 50$ cm$^2$ quadrat into 25 subquadrats of $10 \times 10$ cm$^2$ and then selected 12 subquadrats (8 in the centre and 4 in the middle of the four sides, Figure 1c) to study the quantitative characteristics of pneumatophores. From the 12 selected subquadrats, all pneumatophores taller than 2 cm [33] were counted, and the diameter at ground level ($D_0$) and height (H) were measured. We calculated pneumatophore density (root m$^{-2}$). The total pneumatophore basal area (cm$^2$ m$^{-2}$), total surface area (cm$^2$ m$^{-2}$), and total volume (cm$^3$ m$^{-2}$) were calculated under the assumption that the pneumatophores were cylindrical in shape. A study of the quantitative characteristics of pneumatophores was conducted in the same 12 subquadrats every three months from September 2020 to September 2021.

In October 2020 and November 2021, we collected belowground root samples at an adjacent location approximately 2 m from the $50 \times 50$ cm$^2$ quadrat using soil blocks ($10 \times 10$ cm$^2$) at depths of 0–20 cm. All soil blocks were transported to the laboratory and maintained at a low temperature (7 °C). Subsequently, we separated the pneumatophores and the belowground roots in the soil blocks. Then, the roots were washed through a 0.5 mm sieve and manually separated into live and dead roots based on visual cues (firmness and colour): live roots were firm, pale in colour, or floating on water, and dead roots were broken, brown, or black [34]. The live roots were divided into classes according to their diameters: 0–2 mm (fine root), 2–5 mm (small root), 5–10 mm (medium root), and >10 mm (large root). All root samples were oven-dried at 60 °C and weighed after obtaining a constant weight. The belowground root density was calculated in terms of root mass per unit of soil volume (g m$^{-3}$), and the ratio of pneumatophores and belowground roots was also obtained.

### 2.5. Sedimentation Rates

From the $50 \times 50$ cm$^2$ quadrat used for the pneumatophore study, we selected the centre subquadrat with a size of $10 \times 10$ cm$^2$ to install a sediment trap (Figure 1c). We installed six sediment traps per transect and one on the mudflat. The sediment trap was modified from the filter method [35,36], which used a plastic Petri dish (90 mm in diameter and 10 mm in height) with perforations of approximately 4 mm in diameter for drainage. A pre-weighed 9 cm Whatman qualitative filter was placed in the modified Petri dish, which was covered by an ethylene vinyl acetate (EVA) ring and held by plastic-coated metal straps. All sediment traps were left for three days during the spring tide. The sedimentation rates were studied every month from October 2020 to November 2021. The sedimentation rates were calculated using the following equation:

$$\text{Sedimentation rate} \left( \text{g cm}^{-2} \text{ day}^{-1} \right) = \frac{(t1 - t0)}{3} \tag{1}$$

where t0 and t1 are the preinstallation weight of the oven-dried filter paper and the weight of the filter paper after three days, respectively.

### 2.6. Elevation Changes

The monthly elevation change was recorded on the same day as the sediment trap installation (from October 2020 to November 2021). The sediment pin method [16,37] was used to assess elevation changes of the forest floor along the transects at sampling points 0, 50, and 100 m from the shore and on the mudflat. At the sampling point, an area of $1 \times 1$ m$^2$ was set up to install stainless-steel pins (0.6 cm in diameter and 100 cm in length) at the four corners and one in the centre in September 2020. The stainless-steel pins were vertically inserted into the soil with the remaining 30 cm of their total height above the soil surface. We left the stainless-steel pins for approximately one month to stabilize before we began measuring the height of the pins above the soil surface using a metre ruler (scale 0.1 cm) every month starting in October 2020. Changes in the exposed length of the stainless-steel pins represented elevation changes: a positive elevation change indicates sediment accretion, and a negative change indicates erosion. The net elevation changes of

each $1 \times 1$ m$^2$ area were calculated from a summation of monthly elevation changes and reported as cm per year$^{-1}$.

### 2.7. Environmental Factors

The soil samples were collected at the adjacent location of the sampling points using a soil block at a 0–20 cm depth in October 2020 and November 2021 for soil analyses. The soil samples were analysed for soil particle size distribution using the Hydrometer method [38] and for the contents of soil organic matter using the Walkley–Black chromic acid wet oxidation method [39].

The soil bulk density was measured from the soil samples collected at the same sampling points using a soil core (5-cm diameter) at depths of 0–10 and 0–20 cm in October 2020 and November 2021. All samples were oven-dried at 105 °C and weighed to a constant weight. The soil bulk density was determined from the dry weight of the soil per soil volume (g cm$^{-3}$).

The relative elevation of the forest floor was measured using an instrument siteline builders level (TRACON L5-25, Ushikata Mfg. Co., Ltd., Yokohama, Kanagawa, Japan) every 10 m from a distance of 0–100 m from the shore in September 2020, May 2021, and November 2021.

For the inundation period, we recorded the time during the high tide in which the water approached the datum points along the transects on the shore. We estimated the daily inundation period from the tide table at the Phra Chunlachomklao Fort station (Hydrographic Department, Royal Thai Navy) and obtained the relative elevation. The inundation periods were reported in units of hour day$^{-1}$ from September 2020 to November 2021.

### 2.8. Data Analysis

All statistical analyses were performed by using SPSS Version 22 (IBM Corp., Armonk, NY, USA). The Shapiro–Wilk test was used to determine whether the data were normally distributed. One-way analysis of variance (ANOVA) was performed for data with a normal distribution, and Kruskal–Wallis was used for data with a non-normal distribution for data analysis. The data with significant differences were tested by Tukey's post hoc test (normal distribution) and pairwise comparison (non-normal distribution). The correlation analysis was conducted using Pearson correlation for normally distributed data and Spearman's correlation for non-normally distributed data. The statistical significance of all analyses was accepted at the $p < 0.05$ level.

## 3. Results

### 3.1. Vegetation Structure: Mangrove Trees and Seedlings

We found two mangrove species, *Avicennia alba* Blume and *Rhizophora mucronata* Poir., in the six transects. However, the *A. alba* trees were dominantly distributed throughout the transects (Figure 3) with a relatively high stem density and basal area, with averages of 96% and 99.6%, respectively.

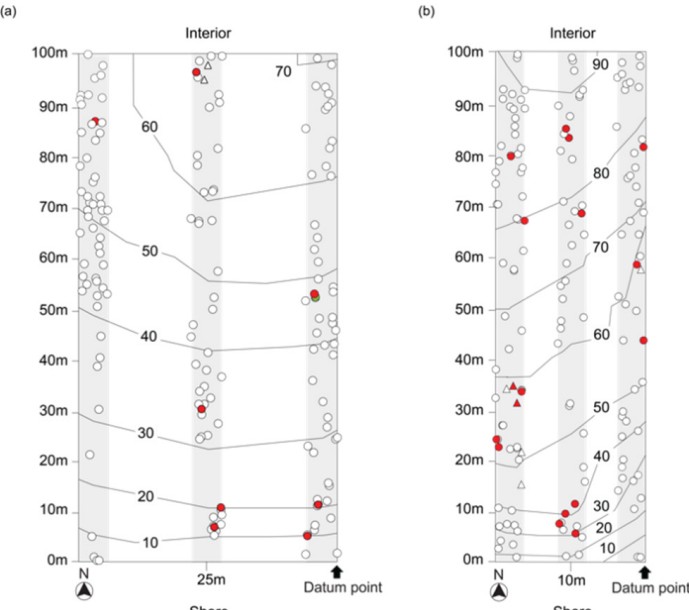

**Figure 3.** Distribution of trees in the six belt transects (grey bands) in 8–9-year-old (**a**) and 10-year-old (**b**) plantations, with the contour lines representing the topographic gradient denoting relative elevation to the datum point; symbols represent mangrove tree species, such as *A. alba* (O) and *R. mucronata* (Δ); the red symbols represent dead trees and the green symbols represent newly recruited trees.

Therefore, we focused on the vegetation structure of abundant *A. alba* trees in the plantation. Along the transect, the total BA of each 10-m interval ranged narrowly from 0.0602 to 0.1207 $m^2$ per 400 $m^{-2}$ (September 2020) and 0.0619 to 0.1129 $m^2$ per 400 $m^{-2}$ (October 2021). The highest total BA and total biomass were found at 60–70 m from the shore with abundant large-DBH trees, while the lowest total BA and total biomass were found at 10–20 m from the shore according to the small- and medium-DBH trees (Table 1 and Figure 4a,b). For the DBH distribution, the trees with small (4.5–10 cm in DBH) and medium (10–20 cm in DBH) DBH values densely occupied the interior area at 50–90 m from the shore (Figure 4). Few newly recruited trees (DBH > 4.5 cm) and dead trees were found from September 2020 to October 2021, as shown in Figure 3.

**Table 1.** Quantitative characteristics of vegetation structure (mean ± SE) across the belt transects in the Bangpu mangrove plantation observed in September 2020 and October 2021.

| Distance from the Shore (m) | Tree Density (Trees 40 $m^{-2}$) | | Total Basal Area ($m^2$ 40 $m^{-2}$) | | Biomass (kg 40 $m^{-2}$) | | | | | |
|---|---|---|---|---|---|---|---|---|---|---|
| | | | | | Aboveground | | Belowground | | Total | |
| | 2020 | 2021 | 2020 | 2021 | 2020 | 2021 | 2020 | 2021 | 2020 | 2021 |
| 0−10 | 6 ± 0.9 | 5 ± 0.8 | 0.0704 | 0.0843 | 411 | 533 | 178 | 221 | 589 | 754 |
| 10−20 | 5 ± 1.3 | 5 ± 1.4 | 0.0602 | 0.0619 | 348 | 371 | 152 | 159 | 500 | 530 |
| 20−30 | 5 ± 1.2 | 4 ± 1.0 | 0.0836 | 0.0910 | 510 | 579 | 216 | 240 | 726 | 819 |
| 30−40 | 4 ± 1.0 | 3 ± 1.0 | 0.0766 | 0.0820 | 471 | 517 | 199 | 216 | 670 | 733 |
| 40−50 | 5 ± 1.0 | 5 ± 1.0 | 0.0809 | 0.0864 | 487 | 530 | 208 | 224 | 695 | 753 |
| 50−60 | 5 ± 1.3 | 5 ± 1.3 | 0.0880 | 0.0966 | 519 | 581 | 224 | 248 | 743 | 829 |
| 60−70 | 6 ± 1.0 | 5 ± 1.1 | 0.1207 | 0.1129 | 761 | 721 | 317 | 298 | 1078 | 1019 |
| 70−80 | 5 ± 1.3 | 5 ± 1.3 | 0.0725 | 0.0770 | 407 | 438 | 180 | 193 | 587 | 631 |
| 80−90 | 6 ± 1.1 | 5 ± 1.0 | 0.0932 | 0.0906 | 539 | 533 | 235 | 230 | 775 | 764 |
| 90−100 | 7 ± 0.5 | 7 ± 0.6 | 0.0983 | 0.1007 | 562 | 581 | 246 | 253 | 809 | 835 |

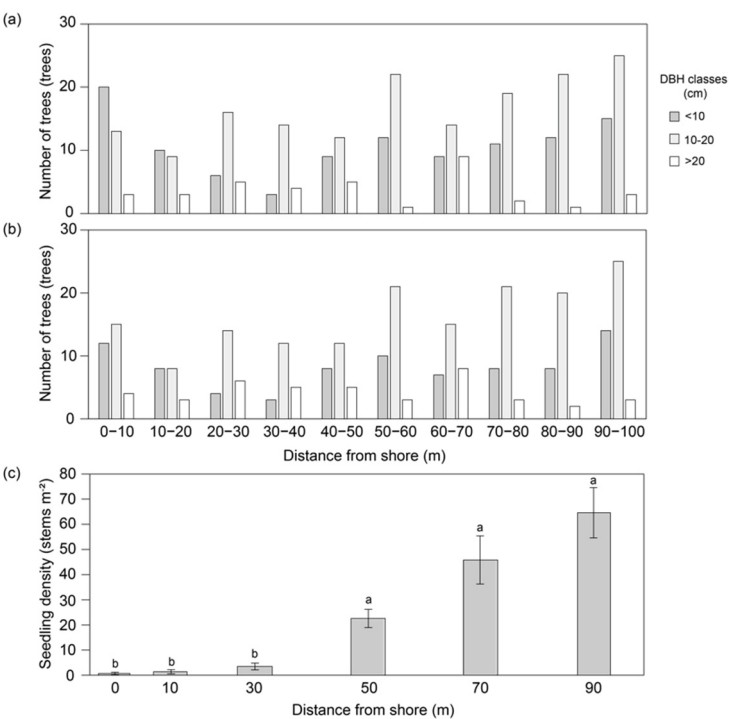

**Figure 4.** Distribution of DBH classes of *A. alba* trees in the belt transects in September 2020 (**a**) and October 2021 (**b**) and the density of *A. alba* seedlings (mean ± SE) across the transects from September 2020 to September 2021 (**c**). Different letters indicate significant differences (Kruskal–Wallis, $p < 0.05$) based on a post hoc test (pairwise comparison).

We found only one species of seedlings, *A. alba*, which had a considerably high density towards the interior areas of all transects. The density of *A. alba* seedlings showed a significant trend according to the distance from the shore: low density at 0–30 m (average of two stems per $m^{-2}$) and high density at distances of 50, 70, and 90 m with an average of 44 stems per $m^{-2}$ (Kruskal–Wallis, H = 86.124, $p < 0.05$, Figure 4c).

### 3.2. Quantitative Characteristics of Pneumatophores and Belowground Roots of A. alba

The quantitative characteristics of pneumatophores did not significantly differ during the study period from September 2020 to November 2021 ($D_0$ and H: Kruskal-Wallis, H = 5.617, $p = 0.153$ and H = 4.797, $p = 0.187$, respectively; density: ANOVA, F = 0.232, $p = 0.874$). Therefore, the quantitative data of pneumatophores of all observation times were compiled and analysed regarding their distance from the shore.

Along the transect, the pneumatophore height and density declined landwards, as shown in Figure 5b,c (Kruskal–Wallis, H = 948.514 and 22.071, respectively, $p < 0.05$). The average height of pneumatophores at 0 m from the shore was 15.4 cm, which was the tallest, and declined to an average of 7.8 cm at 90 m from the shore. Similarly, the total basal area, total volume, and total surface area of pneumatophores decreased landwards, as shown in Figure 5d–f (Kruskal–Wallis, H = 29.940, 74.833, and 74.331, respectively, $p < 0.05$). However, the diameter of pneumatophores ($D_0$) at ground level did not show a clear trend following the distances from the shore (Figure 5a).

Averaging from the six transects, the total belowground root density was 21,854 g $m^{-3}$ in October 2020, which slightly decreased to 20,700 g $m^{-3}$ in November 2021. The total belowground root density ranged from 16,804 to 22,711 g $m^{-3}$, and was not significantly different among the distances along the transect (Figure 6a). Fine roots (<2 mm in diameter) contributed to up to 47% of the total belowground roots and tended to increase landwards, particularly at 50 m onwards (Figure 6a). The other root diameter classes did not vary with the distance from the shore. The mass ratio of pneumatophores to the total belowground roots, which reflects the function of gas exchange supporting belowground roots, was the

highest 0–10 m from the shore, with approximately 23 to 25% of the total roots, while the interior areas were 11 to 14% (Figure 6a).

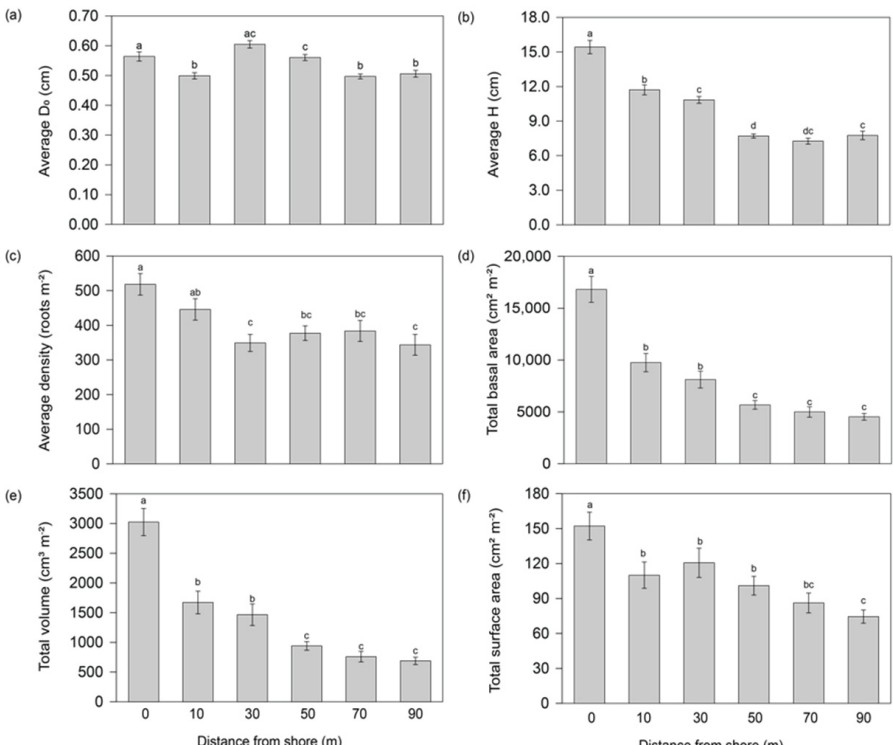

**Figure 5.** Quantitative characteristics of pneumatophores (mean ± SE) along the transects, including diameter at ground level (**a**), height (**b**), density (**c**), total basal area (**d**), total volume (**e**), and total surface area (**f**). Different letters indicate significant differences based on a post hoc test (pairwise comparison) at *p* < 0.05.

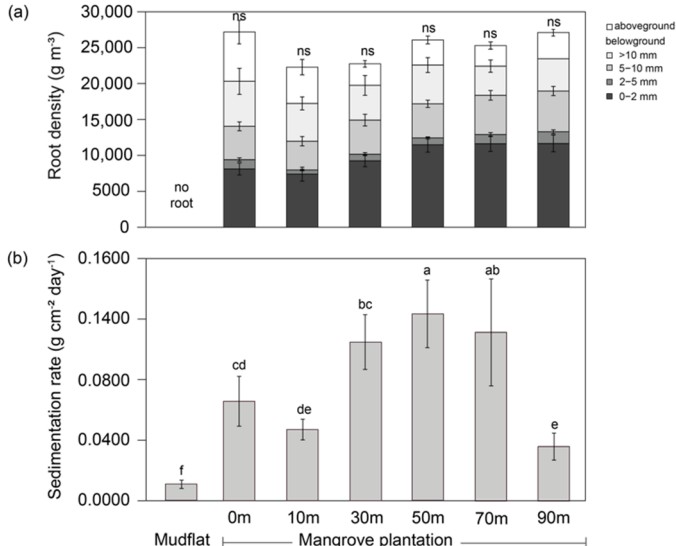

**Figure 6.** Root density (mean ± SE) across the shore (**a**) and average sedimentation rates at the sampling points on the mudflat and along the transect in the mangrove plantation from October 2020 to November 2021 (**b**). The different letters indicate significant differences based on a post hoc test (pairwise comparison) at *p* < 0.05. The designation 'ns' indicates a nonsignificant difference in distance from the shore.

### 3.3. Sedimentation Rates and Elevation Changes

The monthly sedimentation rates during the spring tide fluctuated in a range from 0.0295 to 0.1615 g cm$^{-2}$ day$^{-1}$ over the study period (October 2020 to November 2021). According to the sampling points along the transects, the average sedimentation rates were high at 50 and 70 m from the shore with a maximum of 0.1172 g cm$^{-2}$ day$^{-1}$, and they were low on the mudflat and at 90 m from the shore (Kruskal–Wallis, H = 167.410, $p < 0.01$, Figure 6b).

Monthly elevation changes were investigated from October 2020 to November 2021; we could not obtain data in January and August 2021 due to travel restrictions during the spread of COVID-19.

Monthly elevation changes from October 2020 to November 2021 showed spatiotemporal variations with a wider range on the mudflat (Figure 7a), which had an average range of −0.13–0.09 cm day$^{-1}$. Along the transect, at a distance of 0 m from the shore, the monthly elevation changes had less fluctuations, although there were some negative values, indicating the occurrence of erosion. At distances of 50 m and 100 m from the shore, the mostly positive values of monthly elevation changes indicated sediment accretion (Figure 7a).

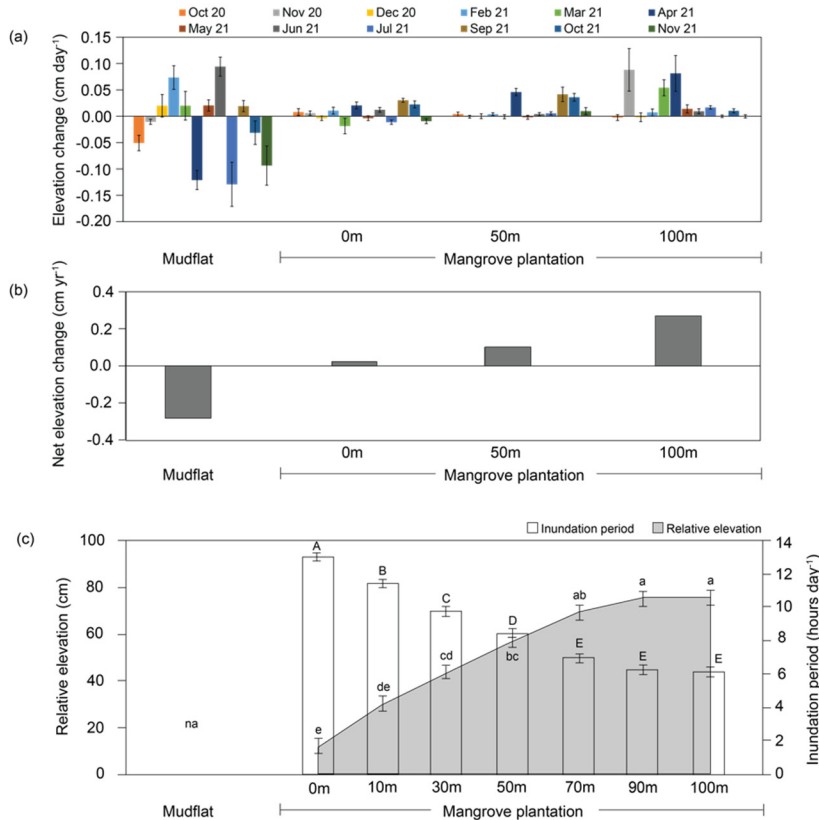

**Figure 7.** Average monthly elevation changes (**a**), net elevation changes (**b**), relative elevation and average inundation period (**c**) on the mudflat and at the distances along the transects from October 2020 to November 2021 (mean ± SE). The bar chart and grey area represent an average inundation period and relative elevation, respectively (**c**). The different letters indicate significant differences based on a post hoc test (pairwise comparison) at $p < 0.05$.

For the one-year period, the net elevation change on the mudflat was calculated at −0.3 cm year$^{-1}$. The net elevation changes of all distances along the transect showed positive values and increased inland (Kruskal–Wallis, H = 19.789, $p < 0.05$, Figure 7b). They were 0.02, 0.1, and 0.3 cm year$^{-1}$ at distances of 0, 50, and 100 m from the shore, respectively.

### 3.4. Environmental Factors

The distribution of soil particle size was dominated by silt followed by sand and clay; the soil was classified as silt to silty loam soil for both study periods. The mud contents (silt and clay particles) were relatively high: over 95% on the mudflat and in the interior areas at 50 to 90 m from the shore (Table 2). The organic matter contents were the highest (2.08–2.39%) at a distance of 0 m from the shore. The soil bulk density was high at distances of 30 to 90 m from the shore and positively correlated with the proportion of sand particles (Spearman, rho = 0.367, $p < 0.05$).

**Table 2.** Soil properties, including bulk density, soil particles, soil texture, and soil organic matter (mean ± SE), in the separate soil depths during October 2020 and November 2021.

| Year | Soil Parameters | Depth (cm) | Distance (m) | | | | | | |
|---|---|---|---|---|---|---|---|---|---|
| | | | Mudflat | 0 | 10 | 30 | 50 | 70 | 90 |
| 2020 | Bulk Density | 0–10 | 1.39 ± 0.05 [ns] | 1.21 ± 0.12 [ns] | 1.39 ± 0.07 [ns] | 1.40 ± 0.07 [ns] | 1.37 ± 0.07 [ns] | 1.30 ± 0.06 [ns] | 1.33 ± 0.08 [ns] |
| | (g cm$^{-3}$) | 0–20 | 1.09 ± 0.07 [bc] | 0.98 ± 0.07 [c] | 1.22 ± 0.08 [ab] | 1.33 ± 0.04 [a] | 1.33 ± 0.03 [a] | 1.32 ± 0.06 [a] | 1.26 ± 0.06 [ab] |
| | Sand (%) | | 0 [bc] | 6.43 [abc] | 10.88 [a] | 7.29 [ab] | 1.70 [abc] | 2.03 [bc] | 0 [c] |
| | Silt (%) | | 89.88 [ns] | 82.60 [ns] | 82.59 [ns] | 86.09 [ns] | 90.35 [ns] | 90.29 [ns] | 89.96 [ns] |
| | Clay (%) | 0–20 | 10.12 [ns] | 10.96 [ns] | 6.53 [ns] | 6.67 [ns] | 7.98 [ns] | 7.67 [ns] | 10.04 [ns] |
| | Soil texture | | Silt | Silty loam | Silty loam | Silty loam | Silt | Silt | Silt |
| | Organic matter (%) | | 1.34 ± 0.14 [ac] | 2.09 ± 0.23 [b] | 1.26 ± 0.18 [cd] | 1.09 ± 0.08 [de] | 0.94 ± 0.09 [e] | 1.19 ± 0.06 [cd] | 1.51 ± 0.11 [ab] |
| 2021 | Bulk density | 0–10 | 1.26 ± 0.06 [ABC] | 1.17 ± 0.05 [C] | 1.33 ± 0.06 [ABC] | 1.43 ± 0.13 [AB] | 1.44 ± 0.07 [A] | 1.43 ± 0.08 [A] | 1.19 ± 0.03 [BC] |
| | (g cm$^{-3}$) | 0–20 | 1.22 ± 0.06 [AB] | 1.03 ± 0.07 [B] | 1.08 ± 0.02 [B] | 1.48 ± 0.08 [A] | 1.23 ± 0.09 [AB] | 1.31 ± 0.08 [A] | 1.19 ± 0.09 [AB] |
| | Sand (%) | | 1.95 [B] | 6.84 [AB] | 13.32 [A] | 11.92 [A] | 8.35 [AB] | 8.02 [AB] | 3.24 [B] |
| | Silt (%) | | 84.27 [ns] | 78.69 [ns] | 78.04 [ns] | 80.93 [ns] | 84.39 [ns] | 83.87 [ns] | 83.92 [ns] |
| | Clay (%) | 0–20 | 13.79 [AB] | 14.47 [AB] | 8.64 [BC] | 7.25 [C] | 7.26 [C] | 8.11 [BC] | 12.84 [A] |
| | Soil texture | | Silty loam | Silty loam | Silty loam | Silt | Silt | Silt | Silt |
| | Organic matter (%) | | 1.56 ± 0.24 [ABC] | 2.38 ± 0.53 [A] | 1.42 ± 0.15 [BC] | 1.22 ± 0.05 [CD] | 1.04 ± 0.08 [D] | 1.29 ± 0.15 [CD] | 1.77 ± 0.17 [AB] |

Remark: Significant differences are shown with different letters ($p < 0.05$) by pairwise comparison (non-parametric test). The letter 'ns' indicates a non-significant difference in terms of the distance from the shore.

The average value of relative elevation ranged from 11 to 76 cm and gradually increased landwards (Kruskal–Wallis, H = 93.237, $p < 0.001$, Figure 7c). The longest period of inundation averaged 13 hours day$^{-1}$ at 0 m from the shore (Figure 7c). It was negatively correlated with the relative elevation (Spearman, rho = −0.631, $p < 0.01$).

### 3.5. Relationships among Mangrove Vegetation Structure, Sedimentation Regimes, and Elevation Changes

The vegetation structure varied across the distance from the shore along the transects. The total BA of the *A. alba* trees and the *A. alba* seedling density positively correlated with the distance from the shore (Spearman, rho = 0.194 and 0.739, respectively, $p < 0.05$). The quantitative characteristics of the pneumatophores, including height, total basal area, total surface area, and total volume, showed negative correlations with the distance from the shore (Table 3). The fine root density positively correlated with the distance from the shore (Spearman, rho = 0.515, $p < 0.05$). The inundation period negatively correlated with the relative elevation of the study plot (Table 3). Moreover, the quantitative characteristics of the pneumatophores, including diameter, height, total basal area, total surface area, and total volume, showed positive correlations with the inundation period (Table 3).

The seedling density showed negative correlations with the total basal area and the total volume of the pneumatophores but showed a positive correlation with the fine root density (Table 3).

The height, total surface area, and total volume of the pneumatophores were negatively correlated with the sedimentation rates (Table 3). However, the total BA and seedling density of *A. alba* did not correlate with the sedimentation rates (Table 3).

**Table 3.** Relationships between mangrove vegetation and sediment across the shore.

| Variable | r/rho | *p*-Value |
|---|---|---|
| **Distance from the shore with:** | | |
| Total BA of trees | 0.194 | 0.035 |
| Seedling density | 0.739 | <0.001 |
| $H_{pneumatophore}$ | −0.817 | <0.001 |
| $BA_{pneumatophore}$ | −0.587 | <0.001 |
| $SA_{pneumatophore}$ | −0.728 | <0.001 |
| $V_{pneumatophore}$ | −0.786 | <0.001 |
| Fine root density | 0.515 | 0.001 |
| **Inundation period with:** | | |
| Relative elevation | −0.631 | <0.001 |
| $D_{0\,pneumatophore}$ | 0.341 | 0.042 |
| $H_{pneumatophore}$ | 0.765 | <0.001 |
| $BA_{pneumatophore}$ | 0.569 | <0.001 |
| $SA_{pneumatophore}$ | 0.660 | <0.001 |
| $V_{pneumatophore}$ | 0.748 | <0.001 |
| **Seedling density with:** | | |
| $BA_{pneumatophore}$ | −0.346 | 0.039 |
| $V_{pneumatophore}$ | −0.578 | <0.001 |
| Fine root density | 0.605 | <0.001 |
| **Sedimentation rate with:** | | |
| $H_{pneumatophore}$ | −0.211 | 0.011 |
| $SA_{pneumatophore}$ | −0.193 | 0.021 |
| $V_{pneumatophore}$ | −0.206 | 0.013 |

Remark: $BA_{pneumatophore}$ is the total basal area of pneumatophores; $SA_{pneumatophore}$ is the total surface area of pneumatophores; $V_{pneumatophore}$ is the total volume of pneumatophores.

## 4. Discussion

Our results demonstrated the potential of the restored vegetation structure of monospecific *A. alba* plantations at Bangpu for protecting coastal areas. Moreover, our findings highlighted that the synergetic roles of mangrove vegetation, including tree stems, seedlings, pneumatophores, and belowground roots, affected sediment accretion in mangrove plantations.

The mangrove plantation at Bangpu showed vegetation structural heterogeneity in that the total BA and tree biomass increased towards the interior areas regarding the high tree density of the trees with a medium DBH (Table 1). According to Jiménez et al. [40], the trees with a small DBH at the forest edge experienced a high mortality rate due to coastal erosion as well as strong waves and wind. In the present study, we found that fewer trees could colonize and reach their maturity in the fringe area of the plantation. Moreover, we found that the density of trees with a medium DBH (10–20 cm) was higher than that of trees with a small DBH (<10 cm), in accordance with the increasing tree diameter after the increasing stand age during the forest development [41,42].

The *Avicennia alba* seedlings were distributed heterogeneously across the transects; the seedling density increased landwards (Figure 4c). Similar spatial distribution patterns of *Avicennia* seedlings were reported in other mangrove forests elsewhere with increasing density towards the interior areas [43,44]. Our study showed that the abundance of trees with a medium or large DBH in the interior (30 to 70 m from the shore) was possibly mother trees providing a source of seeds, implying that the seeds are dispersed and then survive around the mother trees [45]. *Avicennia* seeds are relatively small and highly buoyant, so they can be dispersed by water currents. Therefore, the seeds can be transported into the interior areas and trapped by the structure of the pneumatophores in the forests [46,47]. Thus, the seedlings can establish in the relatively higher area of the intertidal zone [48] with a sufficient period without inundation [45]. Moreover, we found that the seedling density increased toward the interior areas (50–90 m from the shore) which had a high level of sediment accretion (Figure 7b).

The spatial heterogeneity of trees and the seedling distribution were considered to influence the distribution of the *A. alba* roots, both the pneumatophores and the belowground roots. The distribution of pneumatophore characteristics showed an inverse trend with seedling density. These opposite distribution patterns could be a result of competition to occupy the space on the forest floor.

For pneumatophores, the major factors explaining the variation in their quantitative characteristics across the distance from the shore were microtopographic gradients that caused variations in inundation periods. The shoreward areas had a longer inundation period due to the lower relative elevation (Figure 7c). Pneumatophores enhance their volume and surface area under longer inundation conditions to maintain their function in gas exchanges; therefore, the quantitative characteristics of *A. alba* pneumatophores showed a greater density, height, total volume, and total surface area shorewards. However, the pneumatophores tended to grow vertically in height and increase in density rather than diameter to enhance the total volume and surface area (Figure 5). We highlighted that mangroves had plasticity in the quantitative characteristics of pneumatophores in response to environmental changes, as evidenced by our study of the spatial heterogeneity regarding the inundation period.

A similar pattern of spatial variation in which pneumatophore characteristics were also affected by the inundation period was reported in other mangrove species. For *A. marina*, the pneumatophore density and height increased under longer flooding conditions [49]. For *A. germinans*, the pneumatophore density and height were positively related to the inundation period [50]. *Sonneratia* trees had a greater total volume [51] and height [52] in the fringe areas.

Our results showed the spatial variation in *A. alba* pneumatophore characteristics (height, total volume, and total surface area) across the shore inside the plantation, and these characteristics were negatively correlated with the sedimentation rates. Higher sedimentation rates were reported in the middle of the plantation at distances of 30 to 70 m from the shore (Figure 6). The greater quantitative characteristics of pneumatophores (density, height, and total volume) at the fringe caused more intense turbulence [28,52]. The turbulence occurring around the pneumatophores subsequently reduced the opportunity for sediments to settle on the ground, thus the sediments were partly suspended in the water mass and were then transported into the interior areas [53,54]. However, we found the lowest sedimentation rates in the vegetated area at a distance of 90 m from the shore (Figure 6b). We considered that the low sedimentation rates at this distance were caused by the water turbulence due to the existence of a concrete wall (approximately 120 cm in height) located approximately 10 m from the end of the transect that was parallel to the shoreline. Hashim et al. [55] also reported that the adjacent areas to a dyke had lower sediment accumulation than that of the shoreward areas in Malaysian mangrove forests.

In the plantation along the transects, there was no direct relationship between the total BA of the trees or the seedling density and the sedimentation rates. The aboveground structural components of both trees and seedlings might function similarly to the pneumatophores that facilitated the transportation of fine sediments into the interior areas by increasing the turbulence around them. The turbulence in the water mass increased when the seedlings on the forest floor had greater biomass and a complex canopy [56].

We compared the average sedimentation rates on the mudflat ($0.0108 \pm 0.003$ g cm$^{-2}$ day$^{-1}$) with those in the plantation ($0.0815 \pm 0.010$ g cm$^{-2}$ day$^{-1}$). The higher sedimentation rates in the plantation were a result of the presence of aboveground components in the vegetation that reduced the energy of waves and wind [57]. However, Samosorn et al. [37] found that the sedimentation rates on a mudflat ($0.0269$ g cm$^{-2}$ day$^{-1}$) were higher than those in naturally-rehabilitated *A. alba* forests ($0.0196$ g cm$^{-2}$ day$^{-1}$) in Samutsakhon in central Thailand, which had patchy vegetation with a low total BA ($16.5$ m$^{-2}$ ha$^{-1}$). In an estuarine mangrove forest in Vietnam, the sedimentation rates on a mudflat were five to ten times higher than those in very dense vegetation [58]. These results suggest that the

mangrove plantation, which was mature and had an optimal complexity of its aboveground structural components, could enhance the sedimentation rates within the plantation.

The belowground fine roots increased in density towards the interior areas with consistent trends of BA, tree biomass, and seedling abundance. Our results showed a positive relationship between the density of the fine roots and the mud content (silt and clay particles). This finding supports the important roles of fine roots in assisting in sediment binding and stabilization in mangrove forests [21,22,59,60]. A large input of fine root biomass and the necromass of fine roots contribute to the soil carbon pool as organic matter [56], which subsequently enhances sediment accretion in mangrove forests [61].

Our study showed that monthly elevation changes fluctuated highly on the mudflat compared to those observed in the plantation along the transects (Figure 7a). Moreover, the negative values of monthly elevation changes were mostly observed on the mudflat, indicating pronounced erosion. The monthly elevation changes were mostly positive in the plantation (Figure 7a), especially in the most interior area 100 m from the shore. This supported the sediment accretion occurring in the interior areas of the plantation.

Consequently, the net elevation changes on the mudflat implied that erosion occurred at an average rate of $-0.3$ cm yr$^{-1}$ (Figure 7b). The net elevation changes in the plantation showed increasing values that indicated sediment accretion towards the interior areas. Similar to our study, Guo et al. [62] observed net elevation changes in the non-vegetated and vegetated areas of coastal mangrove forests in China and reported average values of $-0.25$ cm yr$^{-1}$ and 1.3 cm yr$^{-1}$, respectively. When net elevation changes were considered within a mangrove forest, Krauss et al. [63], investigating a Micronesian mangrove forest, found that the fringe forest had a lower value, $-0.32$ cm yr$^{-1}$, compared to that of the interior forest, 0.41 cm yr$^{-1}$. However, they did not report any vegetation information associated with elevation changes. Therefore, the vegetation structure of mangrove forests or plantations possibly functions to buffer coastal areas by stabilizing sedimentation processes, reducing erosion, and enhancing sediment accretion.

Our results demonstrated that the vegetation structure, including stems, seedlings, pneumatophores, and belowground roots, had a synergetic function in sedimentation processes. The aboveground structural components with an optimal complexity reduced the impact of waves and facilitated sediment trapping and transport into the interior areas. The belowground components, especially the fine roots, bound and stabilized the fine sediments and enhanced sediment accretion via surface and subsurface sediment accumulation.

For this study, we could not observe pneumatophores monthly due to high tides during the daytime in the early and middle dry season (November to February); instead, we examined pneumatophores every three months, avoiding the months with high tides. Therefore, a temporal change in pneumatophore characteristics did not appear during the whole study period. Our study could only provide information on the relationship between a single type of aerial root (pneumatophore) and local sedimentation regimes in a monospecific plantation in central Thailand. However, the *Rhizophora* tree, which produces prop roots, is one of the mangrove species regularly planted for restoration due to its high survival rate and rapid colonization in estuarine areas [64]. Different types of aerial roots influence sediment accretion, which was reported for *A. marina* with pneumatophores, *Bruguiera gymnorhiza* with knee roots, and *Kandelia candel* with plank roots [65]. Therefore, we suggest further studies of *Rhizophora* plantations that aim to reveal the synergetic roles of the vegetation structure, including the prop roots, on the sediment dynamics.

## 5. Conclusions

We conclude that a monospecific mangrove plantation dominated by *A. alba* potentially stabilized the coastal area through the synergetic roles of the vegetation. An optimal complexity of aboveground structural components reduced the impact of waves and wind and assisted in sediment trapping and transportation. The fine roots, contributed by both trees with medium and large DBHs and high seedling abundance, functioned to bind and

stabilize the fine sediments, leading to sediment accretion in the plantation. We confirmed the roles of fine roots in sedimentation processes indicated by a smaller fluctuation in monthly elevation changes in the plantation compared to that on the mudflat over one year. The net elevation changes indicated that sediment accretion mostly occurred in the plantation, while erosion occurred in the non-vegetated areas of the mudflat. Coastal mangrove forests have been threatened by climate change; thus, we suggest that *A. alba* plantations could maintain their synergetic roles in sedimentation processes due to their behaviour as opportunistic colonizers and the plasticity of their pneumatophores.

**Author Contributions:** Conceptualization, C.R. and S.P.; methodology, C.R. and S.P.; formal analysis, S.H. and C.R.; investigation, S.H., C.R., S.P. and S.U.; writing—original draft preparation, C.R., S.P. and S.H.; writing—review and editing, C.R., S.P., S.U. and V.S.; visualization, S.H.; supervision, C.R. and S.P.; project administration, S.P. and C.R.; funding acquisition, S.P., C.R., S.U. and V.S. All authors have read and agreed to the published version of the manuscript.

**Funding:** This research was funded by Toyota Motor Thailand Co., Ltd. and S.H. was partially funded by the Interdisciplinary Program of Environmental Science, Graduate School Thesis Grant of Chulalongkorn University.

**Data Availability Statement:** The data presented in this study are available on request from the corresponding author.

**Acknowledgments:** This study was fully supported by Toyota Motor Thailand Co., Ltd., for the Corporate Social Responsibility (CSR) program and partially supported by the Interdisciplinary Program of Environmental Science, Graduate School Thesis Grant of Chulalongkorn University to S.H. We thank the staff of the Bangpu Recreation Center for facilitating field data collection. We are grateful to the members of the Plant Ecology Laboratory, Chulalongkorn University, for their assistance with the fieldwork. We appreciate the editors and reviewers.

**Conflicts of Interest:** The authors declare no conflict of interest.

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
