# Peer review of "Synergetic Roles of Mangrove Vegetation on Sediment Accretion in Coastal Mangrove Plantations in Central Thailand"

_forests, doi:10.3390/f13101739_

Round 1
Reviewer 1 Report
The current study is well thought out while being presented and organized in a clear manner. The results are important for the scientific community. I would recommend the manuscript be published after some minor concerns are addressed.
General Comments:
I think in the Introduction, the authors should appreciate the latest advanced work, entitled "Vegetation-induced soil stabilization in coastal area: An example from a natural mangrove forest" (https://doi.org/10.1016/j.catena.2022.106410). I am not one of the authors of the paper.
Specific Comments:
L 16: Add a common name of Avicennia alba, as well.
L 99: Add elevation above sea level of the study area.
L 129: If possible, add the value of the study area's potential (or reference) evapotranspiration.
L 471: At the end of the discussion, add research limitations and future studies.
Author Response
Reviewer 1 (The responses were highlighted in yellow in the revised manuscript.)
Comments and Suggestions for Authors
The current study is well thought out while being presented and organized in a clear manner. The results are important for the scientific community. I would recommend the manuscript be published after some minor concerns are addressed.
- We appreciated your valuable comments and suggestions. The revised manuscript was uploaded to the system and the responses were explained point by point as follows.
General Comments:
Point 1: I think in the Introduction, the authors should appreciate the latest advanced work, entitled "Vegetation-induced soil stabilization in coastal area: An example from a natural mangrove forest" (https://doi.org/10.1016/j.catena.2022.106410). I am not one of the authors of the paper.
Response 1: We edited the introduction as suggested and added the article that the reviewer mentioned in L60–63 as “According to a recent study by Karimi et al. [22] in an Iranian mangrove forest, the higher root length density of marina in the seaward area caused lower soil detachment ratio and erosion compared to the landward area. This suggests that the presence of mangrove roots may help stabilize the shoreline.” We also added the reference to the discussion as mentioned in L490 (reference number 22).
Specific Comments:
Point 2: L 16: Add a common name of Avicennia alba, as well.
Response 2: We could not find a common name in English for alba, so far, the previous research articles on A. alba have never reported its common name. There are several local names reported as Api-api putih (Malay), Api api (Malay), and Samae-Kao (Thai) Therefore, we did not add the common name of A. alba.
Point 3: L 99: Add elevation above sea level of the study area.
Response 3: The elevation of the study area was added in L107 as “The study site is a mangrove plantation in Samut Prakarn Province in central Thailand (13°31′N, 100°39′E), which is located at an elevation of 4 m from the sea level at the Bangpu Recreation Center on the eastern shore of the Chao Phraya River mouth (Figure 1a).”.
Point 4 :L 129: If possible, add the value of the study area's potential (or reference) evapotranspiration.
Response 4: Unfortunately, there has been no report on evapotranspiration at this study site and the adjacent areas in Thailand. The evapotranspiration rates of mangrove forests in Southeast China averaged 2.6 mm day-1 (Liang et al., 2019). However, we cannot apply this value to our study site due to the different dominant tree species and locations of mangrove forests.
Point 5: L 471: At the end of the discussion, add research limitations and future studies.
Response 5: The limitations of the research and future studies were added to the discussion section as suggested in L518–531.
Sincerely,
Chadtip Rodtassana
Reviewer 2 Report
This ms has a sufficient scientific contribution regarding the role of mangroves in sediment accretion, however, more explanation is needed regarding the relationship between tree density (Figures 4a and 4b) and seedling density (Figure 4c) with distance from the coast. Subsequently, quantitative characteristics of pneumatophores (Figure 5) should be explain more what are the factors that cause this trend? It is necessary to explain further regarding the sedimentation rate pattern as presented in Figure 6.
These explanation are very important to conclude as well as answer the overall purposes of this research.

Author Response
Reviewer 2 (The responses were highlighted in blue in the revised manuscript.)
Comments and Suggestions for Authors
Point 1: English language and style are fine/minor spell check is required.
Response 1: We appreciated your constructive comments and suggestions. The revised manuscript was uploaded to the system and the responses were explained point by point. We changed all scientific names into italics style as suggested. However, the scientific names in the original manuscript that we submitted were in italics. There might be some incompatibility of the manuscript files during the file upload and download via the system. We will report this technical problem to the editor.
Point 2: This ms has a sufficient scientific contribution regarding the role of mangroves in sediment accretion, however, more explanation is needed regarding the relationship between tree density (Figures 4a and 4b) and seedling density (Figure 4c) with distance from the coast. Subsequently, quantitative characteristics of pneumatophores (Figure 5) should be explain more what are the factors that cause this trend? It is necessary to explain further regarding the sedimentation rate pattern as presented in Figure 6. These explanation are very important to conclude as well as answer the overall purposes of this research.
Response 2: We addressed the reviewer point by point in detail as follows (please see the text with blue highlight in the revised manuscript):
Point 3: Please be elaborate, why the density of trees with dbh of 10-20cm is mostly higher than trees with dbh of < 10cm?
Response 3: We explained more about the tree density as suggested in L406–412 of the discussion section.
Point 4: Please be elaborate more related the seedling density across distance from shore. What are the factors that cause this trend?
Response 4: The distribution of seedlings and related factors were described; we revised the discussion as suggested in L413–425. More references were added to this part.
Point 5: Should be explain more what are the factors that cause this trend?
Response 5: We revised the discussion to explain more about the trend of pneumatophore characteristics and their related factors as suggested in L431–448. More reference was added to this part.
Point 6: It is necessary to explain further why the sedimentation rate at the landward position of vegetated areas (90m) is the lowest?
Response 6: We explained this in L418–422 in the previous version, but we edited it as suggested in L457–463 with an additional reference.
Point 7: Need an explanation of what is meant by bar charts and gray areas.
Response 7: The content of the chart was already explained in L367–370 in the results. However, we added more explanation as suggested in the legend of Figure 7c as “The bar chart and grey area represent an average inundation period and relative elevation, respectively (Figure 7c).” to make it clearer to the readers.
Sincerely,
Chadtip Rodtassana
Round 2
Reviewer 2 Report
Greats. The responses given are well as well as answers previous input.